# Motion Guidance: Diffusion-Based Image Editing with Differentiable Motion Estimators

**Daniel Geng, Andrew Owens**
University of Michigan
https://dangeng.github.io/motion_guidance

## Abstract

Diffusion models are capable of generating impressive images conditioned on text descriptions, and extensions of these models allow users to edit images at a relatively coarse scale. However, the ability to precisely edit the layout, position, pose, and shape of objects in images with diffusion models is still difficult. To this end, we propose *motion guidance*, a zero-shot technique that allows a user to specify dense, complex motion fields that indicate where each pixel in an image should move. Motion guidance works by steering the diffusion sampling process with the gradients through an off-the-shelf optical flow network. Specifically, we design a guidance loss that encourages the sample to have the desired motion, as estimated by a flow network, while also being visually similar to the source image. By simultaneously sampling from a diffusion model and guiding the sample to have low guidance loss, we can obtain a motion-edited image. We demonstrate that our technique works on complex motions and produces high quality edits of real and generated images.

## 1 Introduction

Recent advances in diffusion models have provided users with the ability to manipulate images in a variety of ways, such as by conditioning on text (Hertz et al., 2022; Tumanyan et al., 2023), instructions (Brooks et al., 2023), or other images (Gal et al., 2022). Yet existing methods often struggle to make many seemingly simple changes to image structure, like moving an object to a specific position or changing its shape.

An emerging line of work has begun to address these problems by incorporating *motion prompts*, such as user-provided displacement vectors that indicate where a given point on an object should move (Chen et al., 2023; Pan et al., 2023a; Shi et al., 2023; Mou et al., 2023; Epstein et al., 2023). However, these methods have significant limitations. First, they are largely limited to *sparse* motion inputs, often only allowing constraints to be placed on one point at a time. This makes it difficult to precisely capture complex motions, especially those that require moving object parts in nonrigid ways. Second, existing methods often require text inputs and place strong restrictions on the underlying network architecture, such as by being restricted to editing objects that receive cross-attention from a text token, by requiring per-image finetuning, or by relying on features from specific layers of specific diffusion architectures.

To address these issues, we propose *motion guidance*, a simple, zero-shot method that takes advantage of powerful, off-the-shelf motion estimators. This method allows users to edit images by specifying a dense, and possibly complex, flow field indicating where each pixel should move in the edited image (Fig. 1 ). We guide the diffusion sampling process using a loss function incorporating an off-the-shelf optical flow estimator, similarly to classifier guidance Dhariwal & Nichol (2021). As part of each diffusion sampling step, we estimate the motion between the generated image and the input source image, and measure the extent to which it deviates from the user-provided flow field. We then augment the noise estimate with gradients through our loss, in the process backpropagating through the optical flow estimator. Simultaneously, we encourage the generated image to be visually similar to the source image by encouraging corresponding pixels to be photoconsistent, allowing our model to trade off between the fidelity of motion and visual appearance.

In contrast to other approaches, our method does not require any training, and does not require a specific diffusion network architecture. We show that our method works for both real and generated

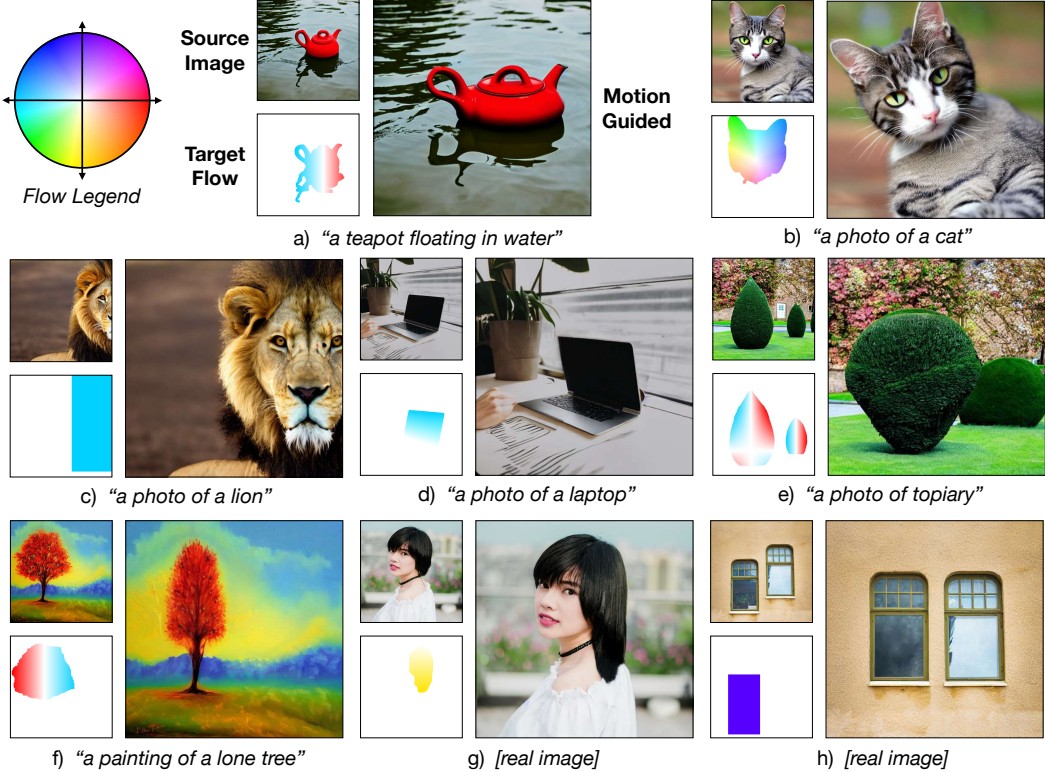

Figure 1: **Flow Guidance.** Given a source image and a target flow, we generate a new image that has the desired flow with respect to the original image. Our method is zero-shot, achieving this by performing guidance through an optical flow network, and works on both real and synthetic images. Note, qualitative results in the main body of this paper were automatically selected for, and random results can be found in Appendix A3.

images, and that it supports a wide range of complex target motions, such as flow fields that have been extracted from a video. We make the following contributions:

- We propose *motion guidance*, a zero-shot technique that allows users to specify a motion field to edit an image.
- We show that off-the-shelf optical flow networks provide a useful guidance signal for diffusion.
- Through qualitative and quantitative experiments, we show that our model can handle a wide range of complex motion fields, including compositions of translations, rotations, homographies, stretching, deformations, and even flow fields extracted from a video.

## 2 RELATED WORK

**Diffusion Models.** Diffusion models (Sohl-Dickstein et al., 2015; Ho et al., 2020; Song et al., 2021) are powerful generative models that learn to reverse a forward process where data is iteratively converted to noise. The reverse process is parameterized by a neural network that estimates the noise, $\epsilon_\theta(\mathbf{x}_t, t, y)$, given the noisy datapoint $\mathbf{x}_t$, the timestep $t$, and optionally some conditioning $y$, for example an embedding of a text prompt. This noise estimate is then used to gradually remove the noise from the data point under various update rules, such as DDPM or DDIM (Song et al., 2020).

**Guidance.** Diffusion models admit a technique called *guidance*, in which the denoising process is perturbed toward a desired outcome. This perturbation may come from the diffusion model itself as in classifier-free guidance (Ho & Salimans, 2022; Nichol et al., 2021), a classifier as in ImageNet classifier guidance Dhariwal & Nichol (2021), or in general the gradients of an energy function. Many forms of guidance function have been proposed including CLIP embedding distance (Wallace et al., 2023; Nichol et al., 2021), LPIPS similarity (Lee et al., 2023), bilateral filters (Gu & Davis, 2023), internal representations of diffusion models themselves (Epstein et al., 2023), and "readout heads" (Luo et al., 2023). Ho et al. (2022) propose to do guidance on a one-step approximation of the clean data, which they term *reconstruction guidance*, and Bansal et al. (2023) apply this

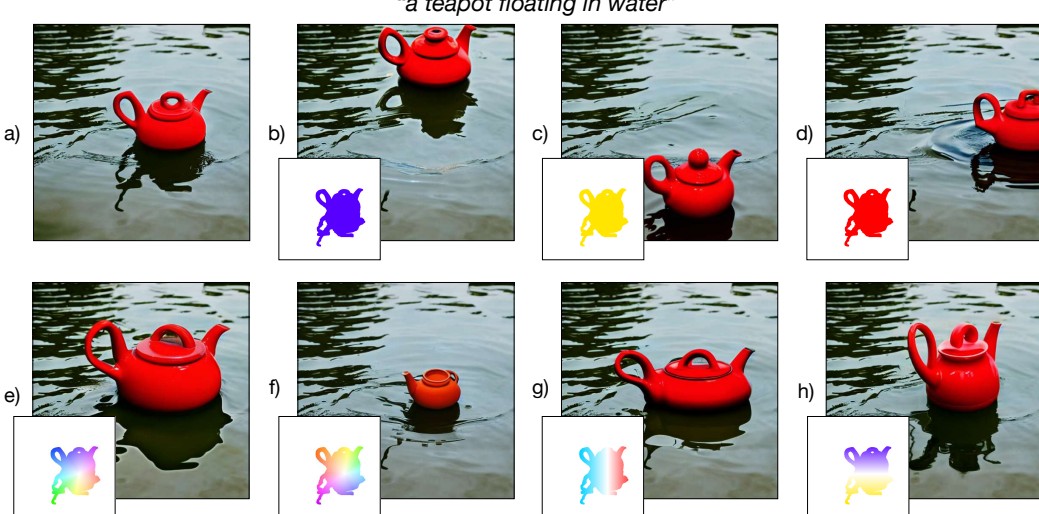

Figure 2: **Moving and Deforming Objects.** We show various motion edits on a single source image (a), demonstrating our method can handle diverse deformations including scaling and stretching. We provide a legend for flow visualization in Figure 1.

idea to achieve guidance on various off-the-shelf models such as segmentation, detection, facial recognition, and style networks. Our method proposes using off-the-shelf optical flow models as guidance to achieve motion-based image editing.

**Image Editing.** Diffusion models, due to their strong image priors, are an attractive starting point for many image editing techniques. Some methods, such as SDEdit (Meng et al., 2022), propose to run the forward diffusion process partially, and then denoise the partially corrupted image. Other methods modify the denoising step as in Lugmayr et al. (2022) and Wang et al. (2023). Still others finetune a diffusion model for image manipulation tasks (Brooks et al., 2023; Zhang & Agrawala, 2023). Another class of diffusion-based image manipulation techniques uses the features and attention inside the denoising models themselves to edit images. These techniques work due to the observation that features can encode attributes, identity, style, and structure. Methods in this vein include Prompt-to-Prompt (Hertz et al., 2022), Plug-and-Play (Tumanyan et al., 2023), and Self-Guidance (Epstein et al., 2023).

The majority of these techniques enable impressive editing of image style, but are unable to modify the image structure. A more recent line of work seeks to achieve editing of structure. These methods are closest to ours. DragGAN (Pan et al., 2023a) proposes an optimization scheme over StyleGAN features that allows users to "drag" points in an image to new locations. However, these results are constrained to narrowly trained GANs. In order to achieve much wider applicability concurrent work has adapted the DragGAN optimization scheme to diffusion models (Shi et al., 2023; Mou et al., 2023), however these approaches either require fine-tuning with LoRA (Hu et al., 2021) for each image or have not been shown to work on complex, densely defined deformations. In recent work, Epstein et al. (2023) perform motion editing with classifier guidance, using the activations and textual-visual cross-attention of the diffusion model to specify object position and shape. This limits the complexity of deformations and constrains manipulable objects to those that have a corresponding textual token in the prompt. In contrast, our guidance comes from an off-the-shelf optical flow network and thus does not suffer from these problems, nor does it rely on a particular architecture for the diffusion model. Overall our method seeks to build upon past and concurrent work, by enabling dense, complex, pixel-level motion manipulation for a wide range of images in a zero-shot manner.

## 3 METHOD

Our goal is to allow a user to edit a source image $\mathbf{x}^* \in \mathbb{R}^{w \times h \times 3}$ by specifying a target *flow field* $\mathbf{f} \in \mathbb{R}^{w \times h \times 2}$ that indicates how each pixel should move. To do this, we develop techniques that guide a diffusion model's denoising process using an off-the-shelf optical flow network.

## 3.1 GUIDANCE IN DIFFUSION MODELS

Diffusion models (Sohl-Dickstein et al., 2015; Song et al., 2021) generate images by repeatedly denoising a sample of random noise. They are commonly represented (Ho et al., 2020) as functions $\epsilon_\theta(\mathbf{x}_t; t, y)$ that predict the noise in a noisy data point $\mathbf{x}_t$ at step $t$ of the diffusion process, using an optional conditioning signal $y$. Classifier guidance (Dhariwal & Nichol, 2021) seeks to generate samples that minimize a function $L(\mathbf{x})$, such as an object classification loss, by augmenting the noise estimate with the gradients of $L$:

$$\tilde{\epsilon}_\theta(\mathbf{x}_t; t, y) = \epsilon_\theta(\mathbf{x}_t; t, y) + \sigma_t \nabla_{\mathbf{x}_t} L(\mathbf{x}_t), \tag{1}$$

where $\tilde{\epsilon}_\theta$ is the new noise function and $\sigma_t$ is a weighting schedule. A key benefit of this approach is that it does not require retraining, and it does not explicitly place restrictions on $L$, except for differentiability. Recent work has shown that this approach can be straightforwardly extended to use functions $L$ that accept only clean (denoised) images as input (Bansal et al., 2023).

## 3.2 MOTION GUIDANCE

We will use guidance to manipulate the positions and shapes of objects in images. We design a guidance function $L(\mathbf{x})$ (Eq. 1) that measures how well a generated image, $\mathbf{x}$, captures the desired motion. We encourage the *optical flow* between $\mathbf{x}^*$ and $\mathbf{x}$ to be $\mathbf{f}$. Given a (differentiable) off-the-shelf optical flow estimator $F(\cdot, \cdot)$, we will minimize the loss

$$L_{\text{flow}}(\mathbf{x}) = \|F(\mathbf{x}^*, \mathbf{x}) - \mathbf{f}\|_1. \tag{2}$$

While we found that performing guidance on this loss could produce the desired motions, we also observed that the edited objects would often change color or texture. This is because motion estimation models are invariant to many color variations[1], thus there is little to constrain the appearance of the object to match that of the source image. Inspired by Pan et al. (2023b) and Geng et al. (2022), to address this we add a loss that encourages corresponding pixels to have similar colors

$$L_{\text{color}}(\mathbf{x}) = \|\mathbf{x}^* - \text{warp}(\mathbf{x}, F(\mathbf{x}^*, \mathbf{x}))\|_1, \tag{3}$$

where $\text{warp}(\mathbf{x}, \mathbf{f})$ indicates a backward warp of an image $\mathbf{x}$ using the displacement field $\mathbf{f}$. The full loss we use for guidance is then

$$L(\mathbf{x}) = \lambda_{\text{flow}} L_{\text{flow}}(\mathbf{x}) + \lambda_{\text{color}} \mathbf{m}_{\text{color}} L_{\text{color}}(\mathbf{x}), \tag{4}$$

where $\lambda_{\text{flow}}$ and $\lambda_{\text{color}}$ are hyperparameters that trade off between the model's adherence to the target motion and the visual fidelity to objects in the source image, and $\mathbf{m}_{\text{color}}$ is a mask to handle occlusions, explained below.

## 3.3 IMPLEMENTING MOTION GUIDANCE

We find a number of techniques useful for producing high quality edits with motion guidance.

**Handling Occlusions.** When objects move, they occlude pixels in the background. Since these background pixels have no correspondence in the generated image, the color loss (Eq. 3) is counterproductive. Similar to Li et al. (2023), we adopt the convention that the target flow specifies the motions of the *foreground*, i.e., that when an object moves, it will occlude the pixels that it overlaps with. To implement this, we create an occlusion mask from the target flow and mask out the color loss in the occluded regions. Please see Appendix A1 for details.

**Edit Mask.** For a given target flow, often many pixels will not move at all. In these cases, we can reuse content from the source image by automatically constructing an *edit mask* $\mathbf{m}$ from the target flow, indicating which pixels require editing. This mask consists of all locations that any pixel is moving to or from. To apply the edit mask during diffusion sampling, we assume access to a sequence of noisy versions of the source image $\mathbf{x}^*$ for each timestep, which we denote $\mathbf{x}_t^*$. For real images this sequence can be constructed by injecting the appropriate amount of noise into $\mathbf{x}^*$ at each timestep $t$, or alternatively it can be obtained through DDIM inversion (Song et al., 2020). For images sampled from the diffusion model, we can also cache the $\mathbf{x}_t^*$ from the reverse process. Then, similarly to Song et al. (2021) and Lugmayr et al. (2022), during our denoising process at each timestep $t$ we replace pixels outside of the edit mask with the cached content: $\mathbf{x}_t \leftarrow \mathbf{m} \odot \mathbf{x}_t + (1 - \mathbf{m}) \odot \mathbf{x}_t^*$.

---

[1]This is quite useful for flow estimation on videos, where lighting may change rapidly frame to frame.

**Handling Noisy Images.** In standard classifier guidance, noisy images $\mathbf{x}_t$ are passed into the guidance function. In our case, this results in a distribution mismatch with the off-the-shelf optical flow model, which has only been trained on clean images. To address this we adopt the technique of Bansal et al. (2023) and Ho et al. (2022), and compute the guidance function on a *one step* approximation of the clean $\mathbf{x}_0$ given by

$$\hat{\mathbf{x}}_0 = \frac{\mathbf{x}_t - \sqrt{1 - \alpha_t}\epsilon_\theta(\mathbf{x}_t, t)}{\sqrt{\alpha_t}}, \tag{5}$$

resulting in the gradient $\nabla_{\mathbf{x}_t}\mathcal{L}(\hat{\mathbf{x}}_0(\mathbf{x}_t))$, with a more in-domain input to the guidance function.

**Recursive Denoising.** Another problem we encounter is that the optical flow model, due to its complexity and size, can be quite hard to optimize through guidance. Previous work (Lugmayr et al., 2022; Wang et al., 2023; Bansal et al., 2023) has shown that repeating each denoising step for a total of $K$ steps, where $K$ is some hyperparameter, can result in much better convergence. We adopt this *recursive denoising* technique to stabilize our method and empirically find that this resolves many optimization instabilities, in part due to an iterative refinement effect (Lugmayr et al., 2022) and also in part due to the additional steps that the guidance process takes, effectively resulting in more optimization steps over the guidance energy.

**Guidance Clipping.** We also find that clipping guidance gradients prevents instabilities during denoising. Concurrent work by Gu & Davis (2023) introduces an adaptive clipping strategy, but we find that simply clipping gradients by their norm to a pre-set threshold, $c_g$, works well in our case.

## 4 RESULTS

We evaluate our method's ability to manipulate image structure, both qualitatively and quantitatively, on real and generated images. Additional results can be found in Appendix A3.

### 4.1 IMPLEMENTATION DETAILS

We use RAFT (Teed & Deng, 2020) as our flow model. To create target flow fields, we compose a set of elementary flows and use a segmentation model (Kirillov et al., 2023) for masking. Target flow construction and hyperparameters are discussed in detail in Appendix A1. We additionally implement a simple GUI to generate various flow fields, which we describe in Appendix Figure A1 and A2. For our experiments we use Stable Diffusion (Rombach et al., 2021). Rather than performing diffusion directly on pixels, Stable Diffusion performs diffusion in a latent space, with an encoder and decoder to convert between pixel and latent space. To accommodate this, we precompose the decoder with the motion guidance function, $L(\mathcal{D}(\cdot))$, so that the guidance function can accept latent codes. Additionally, we downsample our edit mask to $64 \times 64$, the spatial size of the Stable Diffusion latent space.

### 4.2 QUALITATIVE RESULTS

Main qualitative results can be found in Figures 1, 2, 3, and 4. Following previous work (Ramesh et al., 2021; Bansal et al., 2023) we generate multiple samples per example and choose the best result according to the guidance energy. Details can be found in Sec. A1, and random and top-5 qualitative results can be found in Sec. A3. We point out several strengths of our method:

**Disocclusions.** Because our method is built on top of a powerful diffusion model, it is able to reason about disoccluded regions. For example, in Figure 1c our model moves the lion to the left and is able to fill in the previously occluded face.

**Diverse Flows.** Our method successfully manipulates images using translations (Figures 1c, 1h, 3a), rotations (Figures 1b, 4a), stretches (Figures 1a, 1f, 1g), and scaling (Figures 2e, 2f, 4b). It also handles complex deformations (Figures 1e, 3c, 4c), homographies (Figures 1d, 3b), and multiple objects (Figures 4e, 1e). Moreover, our method is able to handle relatively large movements (Figures 2b, 2c, 2d, 3a). We are able to achieve successes on diverse flows due to the generality of both the optical flow and diffusion model we use.

**Diverse Images.**   We show that our approach works on a wide range of input images with diverse objects and backgrounds, including non-photorealistic images such as paintings or sketches (Figures 1f , 4b , 5a , 5b , 5c ), despite the fact that the optical flow network was not trained on these styles. We attribute this success due to the fact that optical flow is a relatively low level signal, and does not require understanding of high level semantics to estimate. Additionally, we show that our method works for both synthetic images as well as real images (Figures 1g , 1h , 3c , 4e , 7b , 7c , 7d ).

**Soft Optimization.**   In contrast to forward warping, our method optimizes a soft constraint: satisfy the flow while also producing a plausible image. Because of this, our method works even for coarsely defined target flows as shown in Figures 1c , 1h , 5b . This is particularly useful when we do *motion transfer* in Section 4.6 and Figure 7 , where the extracted flow is only roughly aligned with the source image. Another benefit of soft optimization is that our method works even when target flows are ambiguous, such as when flows specify that two objects should overlap as in Figure 1e .

**Text Conditioning.**   Because our method is based on guidance, it is independent of text conditioning. This is useful for editing real images, for which no caption exists. In all figures, samples conditioned on text will have the prompt written in italics below the example, and real images are labeled as "*[real image]*" and are unconditionally generated.

## 4.3   ABLATIONS

In Figure 3 we qualitatively ablate out key components of our guidance function.

**No Recursive Denoising.**   Without recursive denoising our method converges much less frequently, often resulting in samples with heavy artifacts as in Figure 3c . Even in cases where the guidance mostly succeeds, artifacts can still occur as in Figure 3a .

**No Color Loss.**   Without the color loss, our method generally moves objects to the correct locations, but objects tend to change color as in Figure 3a . These samples still achieve a low flow loss because of the color invariance of the optical flow network, highlighting the need for the color loss. In some cases we see catastrophic failure, as in Figure 3c .

**No Flow Loss.**   Removing the flow loss also removes any knowledge the method has of the target flow. To address this we also modify the color loss (Eq. 3 ), replacing the computed flow with the target flow. Without the flow loss our method is able to move objects to the correct location but often hallucinates things in disoccluded areas, such as the half apple in Figure 3a or the waterfall in Figure 3c . This is because the color loss does not produce any gradient signal in disoccluded areas due to the warping operation. In the disoccluded areas the diffusion model effectively acts freely with no guidance at all, resulting in these hallucinations.

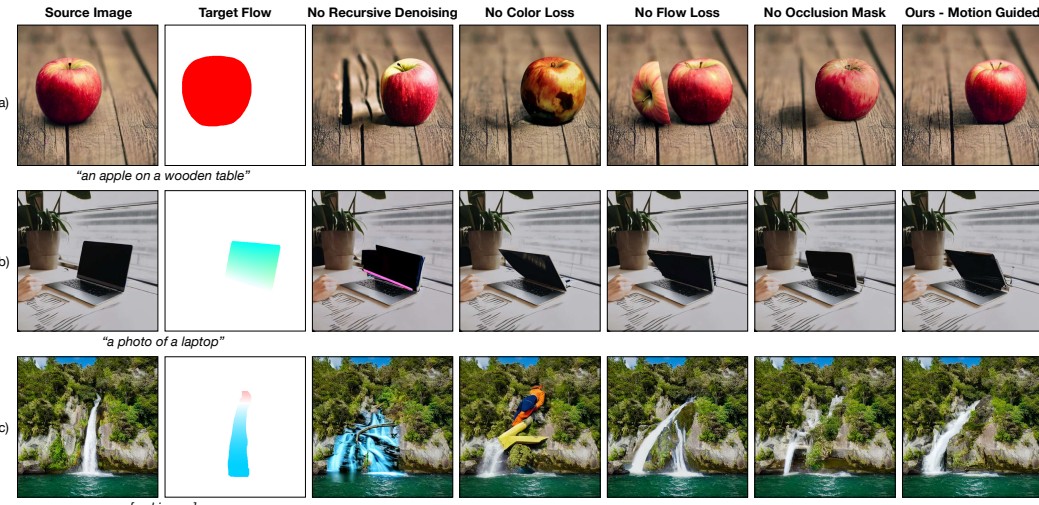

Figure 3: **Ablations.** We qualitatively ablate out techniques we use to achieve motion guidance. For a discussion please see Section 4.3 . We provide a legend for the flow visualization in Figure 1.

**No Occlusion Masking.** Without occlusion masking the color loss is incorrect in occluded areas. This is because the target flow in occluded areas is invalid, so the warping operation mismatches pixels. As a result we tend to see "ghosting" of moved objects, where objects blend into the background or take on the color of the disoccluded background, as can be seen in Figure 3a or Figure 3c. In more extreme cases, the object can fail to move into the occluding region at all, as in Figure 3b.

## 4.4 BASELINES

We present comparisons between our method and baselines in Figure 4. For a more extensive set of comparisons please see Figure A7 in the appendix.

**InstructPix2Pix.** InstructPix2Pix (Brooks et al., 2023) distills Prompt-to-Prompt (Hertz et al., 2022) into a diffusion model that is conditioned on textual editing instructions. Because this model is only text-conditioned, it is not possible to give it access to the target flow, but we try to faithfully summarize the flow in an instruction. As can be seen, despite our best efforts, InstructPix2Pix is never able to move objects significantly although it does occasionally alter the image in rather unpredictable ways. This failure can be attributed to two factors. Firstly, text is not an ideal method for describing motion. For example it is quite hard to encapsulate the motion in Figure 4e in text without being excessively verbose. And secondly, feature-copying methods such as Prompt-to-Prompt often fail to make significant structural edits to images.

**Forward Warp with SDEdit.** We also use baselines that explicitly use the target flow. Specifically, we forward warp the latent code by the target flow and use SDEdit (Meng et al., 2022), conditioned on the text prompt (if given) at various noise levels to "clean" the warped image. While SDEdit succeeds to a degree in Figure 4b and Figure 4d, the results are of lower quality and the failure cases contain severe errors.

**Forward Warp with RePaint.** In addition to cleaning the forward warped images using SDEdit, we try the diffusion based inpainting method RePaint (Lugmayr et al., 2022), conditioned on the text prompt (if given), to fill in disoccluded areas in the forward warp. The inpainted areas are generally realistic looking, but often not exactly correct. For example in Figure 4d, an additional teapot is generated in the disoccluded area.

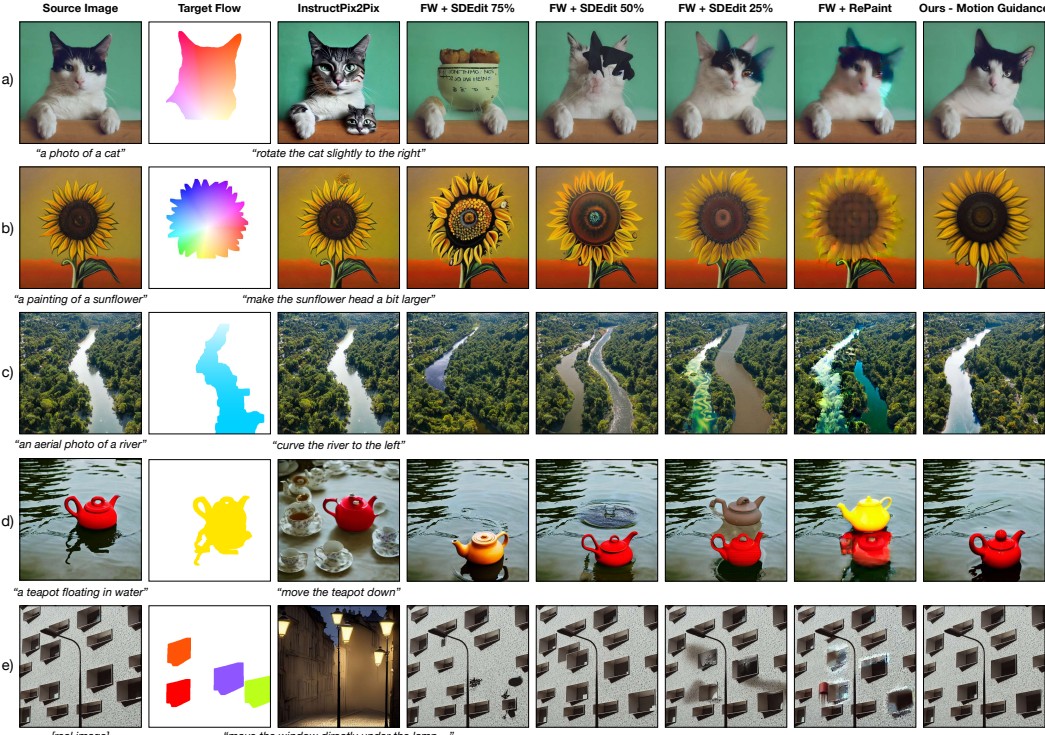

Figure 4: **Baselines.** We show qualitative examples from various baselines and our method. The instruction used for InstructPix2Pix is shown beneath each InstructPix2Pix sample. For a discussion please see Section 4.4. We provide a legend for the flow visualization in Figure 1.

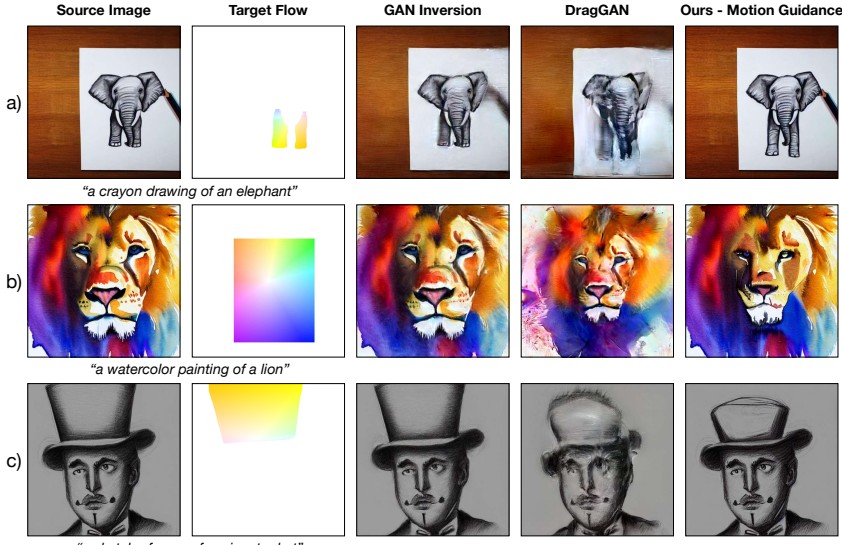

| Source Image | Target Flow | GAN Inversion | DragGAN | Ours - Motion Guidance |

a) *"a crayon drawing of an elephant"*

b) *"a watercolor painting of a lion"*

c) *"a sketch of a man face in a tophat"*

Figure 5: **Comparison to DragGAN.** DragGAN works only on domains for which a StyleGAN has been trained on. Attempts to edit real images that are out-of-domain, even if they are invertible, results in failures that our model handles well. Here we show results on StyleGANs trained for (a) elephants (b) lions (c) faces. We provide a legend for the flow visualization in Figure 1.

**DragGAN.** Recent work has proposed an iterative two-step optimization procedure over GAN features to "drag" user specified handle points to target points. While the method, DragGAN (Pan et al., 2023a), is impressive it has only been demonstrated on StyleGANs (Karras et al., 2020) trained on narrow domains, such as dogs, lions, or human faces, limiting the scope of such a method. To demonstrate this, we show out-of-domain cases in which DragGAN fails in Figure 5. Following DragGAN, to invert an image to a StyleGAN latent code we use PTI (Roich et al., 2021). We then subsample the target flow to obtain handle and target points that are required by DragGAN. Drag-GAN GPU memory usage scales linearly with the number of handle points, so densely sampling the flow is not feasible, even on an NVIDIA A40 GPU with 48 GB of memory. DragGAN also supports an edit mask, so we provide it with the same automatically generated edit mask that our method uses. We find that while the GAN inversion is successful even on out-of-domain cases, DragGAN motion edits contain many artifacts.

## 4.5 QUANTITATIVE RESULTS

We are interested in not just optimizing for a specific target flow, but also generating images that are faithful to the source image and that are coherent. Therefore we evaluate our method on two metrics which are designed to reflect the trade-offs we are interested in. One metric is the **Flow Loss**, $\mathcal{L}_{flow}$ from Equation 2, which should measure how accurately a generated image adheres to the target flow. Because our guidance function actually uses $\mathcal{L}_{flow}$ with RAFT, we present results using both RAFT and GMFlow (Xu et al., 2022) in case we are overfitting. For our second metric we compute the **CLIP Similarity** as the cosine distance between the CLIP visual embeddings of the source and generated images. We treat this as a measure of faithfulness to the source image as well as a measure of general image quality.

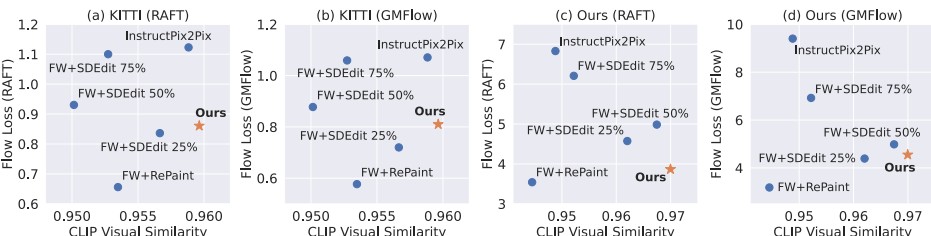

Figure 6: **Quantitative Metrics.** We show performance of baselines and our method on the Flow Loss and CLIP Similarity metrics, on two different datasets, *Our* curated dataset and an automatically generated *KITTI* dataset. Please see Section 4.5 for further discussion.

We evaluate on two different datasets. The first dataset is composed of examples with handcrafted target flows, a subset of which can be seen in Figures 1 , 2 , 3 , 4 , and 7 . This dataset has the advantage of containing interesting motions that are of practical interest. In addition, we can write highly specific instructions for the InstructPix2Pix baseline for a fair comparison. However, this dataset is curated to an extent. We ameliorate this by performing an additional evaluation on an automatically generated dataset based on KITTI (Geiger et al., 2012), which contains egocentric driving videos with labeled bounding boxes on cars. To build our dataset we construct target flows consisting of random translations on cars. For more details on the datasets please see Appendix A2 .

We show results in Figure 6 . As can be seen, our method offers an attractive trade-off between satisfying the target flow and keeping faithful to the source image. The RePaint baseline, which is essentially forward warping and then inpainting, achieves a low flow loss due to the explicit warp operation but suffers from artifacts due to aliasing and inpainting and thereby results in a low CLIP similarity. The SDEdit baseline, which also uses forward warping, can achieve a low flow loss if the noise level is low, but also falls short on the visual similarity metric.

## 4.6 MOTION TRANSFER

Sometimes it is hard to specify a flow by hand. In these cases we find that our method can successfully "transfer" motion from a video to an image. We give examples in Figure 7 , where we extract the flow from a video of the earth rotating and use it as our target flow. We find that even if the extracted flow does not overlap perfectly with the target image the desired motion can be achieved. We attribute this to the fact that our guidance is a *soft* optimization and the diffusion model overall ensures high-level semantic consistency.

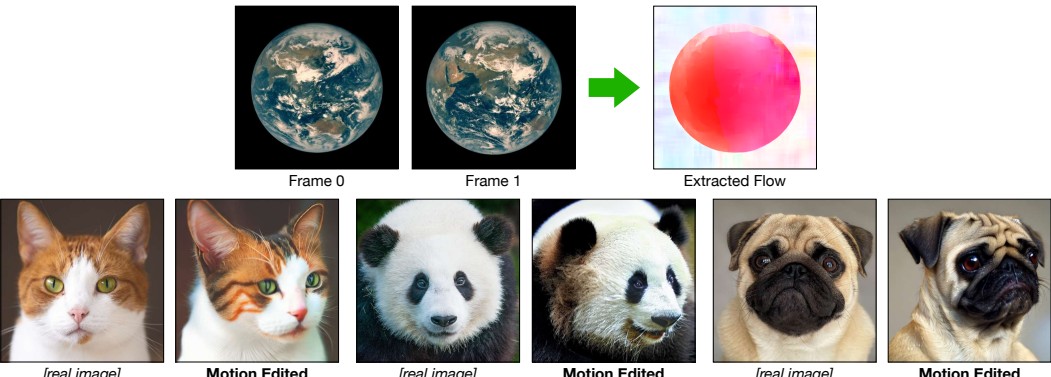

Figure 7: **Motion Transfer.** Our method can use the motion extracted from a source video to motion-edit a completely different image. Note we use fewer recursive denoising steps for these samples than for other samples because we found it to improve results slightly. We provide a legend for the flow visualization in Figure 1.

## 5 DISCUSSION

**Limitations.** While our method can produce high quality motion-conditioned edits, it is also susceptible to various weaknesses. In particular, we inherit the deficiencies of diffusion models and guidance based methods, such as slow sampling speed. The use of universal guidance also introduces instability in the sampling process, which is only partially addressed by the recursive denoising strategy (Sec. 3.3). In addition, we also inherit the limitations of our optical flow method, and find that certain target flows are not possible. We give a comprehensive discussion of our limitations in Appendix A4 .

### 5.1 CONCLUSION

Our results suggest that we can manipulate the positions and shapes of objects in images by guiding the diffusion process using an optical flow estimator. Our proposed method is simple, does not require text or rely on a particular architecture, and does not require any training. We see our work as a step in two research directions. The first direction is in integrating differentiable motion estimation models into image manipulation models. Second, our work opens the possibility of repurposing other low-level computer vision models for image generation tasks through diffusion guidance.

## ACKNOWLEDGEMENTS

This project was supported by a Sony Research Award. Daniel Geng is supported by a National Science Foundation Graduate Research Fellowship under Grant No. 1841052. Daniel would also like to thank Berlin for a wonderful summer, during which this research was conducted.

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

## A1   IMPLEMENTATION DETAILS

**Hyperparameters**   We use Stable Diffusion v1.4 with a DDIM sampler for 500 steps, and we generate images at a resolution of $512 \times 512$. All experiments are conducted on a single NVIDIA A40 GPU. For our motion guidance function (Eq. 4) we found that setting $\lambda_{\text{color}}$ to 100 and $\lambda_{\text{flow}}$ to 3 worked well. In addition, in our implementation we scale the guidance gradients by a global weight of 300. We set the gradient clipping threshold $c_g$ to be 200 and take $K = 10$ recursive denoising steps. For just the motion transfer results we set $K = 10, 5, 2$ for the cat, pug, and panda respectively. For the RAFT optical flow model we use the public checkpoint trained on FlyingChairs (Dosovitskiy et al., 2015) and FlyingThings3D (Mayer et al., 2016) with 5 iterative updates.

**Guidance Schedule**   Previous work (Saharia et al., 2022; Epstein et al., 2023; Li et al., 2022) has shown that sample quality can improve with well-chosen guidance schedules, that is, applying guidance with varying strength through out denoising. We find that applying guidance as normal for the first 400 steps and turning it completely off for the final 100 steps is beneficial in our case and we use this schedule for all samples. Intuitively, in the final 100 steps the low-level structure of the image is already sampled and little can be done to alter the motion of the sample. Turning off guidance during this phase allows the model to focus on synthesizing realistic, high quality details.

**Occlusion Masking**   In order to create an occlusion mask we need additional information or assumptions beyond just the flow. For example, depth could indicate to us which objects would occlude others. Similar to Li et al. (2023), in the absence of this information we make the simplifying assumption that the target flow is applied to a foreground object. Then in order to construct an occlusion mask we find all background regions that will be overwritten by the target flow by performing a warp and add these pixels to the occlusion mask, which is used to mask the color loss.

**Reranking**   Following previous work (Ramesh et al., 2021; Bansal et al., 2023) we generate multiple samples and automatically filter these samples. Specifically, we generate 32 samples per example and re-rank based on the final motion guidance loss value. We find that re-ranking is particularly useful for filtering out poorly converged samples, which we discuss in more detail in Appendix A4. We present both top-5 and random samples in Appendix A3. Note we do not use re-ranking for our quantitative evaluations in Section 4.5.

**Target Flow Construction**   We craft target optical flows by composing translations, rotations, scaling, stretching, and shearing. For more complex flows, such as with the river or the topiary samples, we interpolate between discrete displacements with sine functions. For homographies, as in the laptop example, we define starting and ending locations for corners and fit a homography $\mathbf{H}$ to these points. We then extract a flow by computing $\mathbf{Hx} - \mathbf{x}$, the displacement for each point $\mathbf{x}$ under the homography.

In most cases we mask out the flow using a mask extracted using SAM (Kirillov et al., 2023). This mask is dilated by a few pixels by applying an all ones convolution, equivalent to a box blur, and then thresholding. We find this helpful because SAM often slightly under estimates the extent of an object. All flow outside of the dilated mask is set to zero. We do not mask for the motion transfer samples.

For ease of use, we create a simple GUI that combines the segmentation step and the generation of optical flows. Our GUI allows users to create translations, rotations, stretches, scaling, and complex deformations simply by clicking a dragging. For more details and examples, see Figures A1 and A2.

## A2   DATASET DETAILS

For the dataset based on KITTI we use frames with bounding box car annotations. We first filter out frames with no cars, frames with cars that are too small or too large, and frames with occluded cars. Occlusion information is annotated by the dataset. For each frame we choose a car and crop the frame to that car, and resize to obtain a $512 \times 512$ image. We then construct a flow representing a translation of the bounding box, with a uniformly random direction and a magnitude uniformly sampled between 10 and 50 pixels. We use a total of 226 examples. For the InstructPix2Pix baseline we automatically construct an instruction prompt by mapping the randomly sampled direction to one of 8 sentences such as "move the car to the left" or "move the car to the right and down." We visualize results of motion guidance on this dataset in Figure A3.

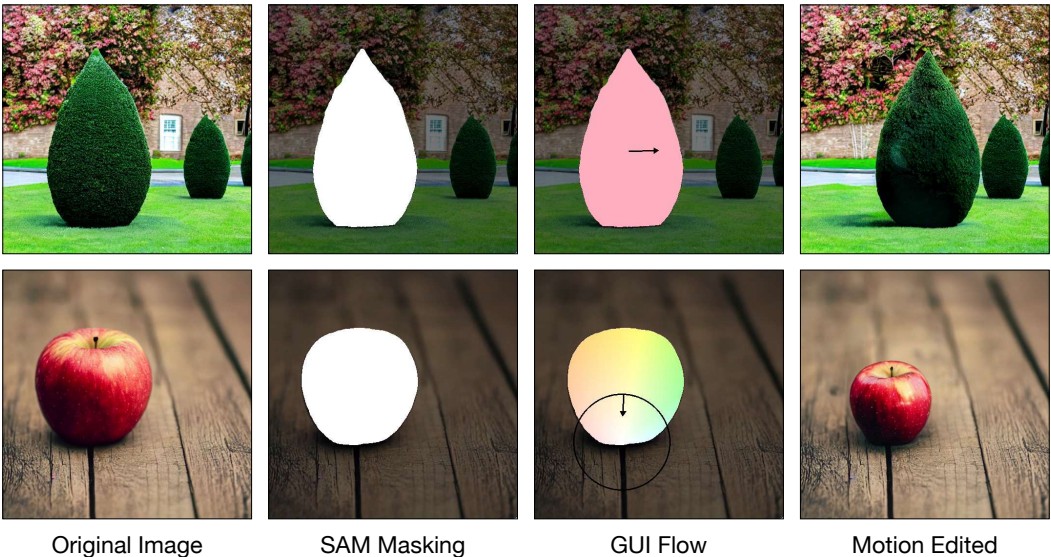

| Original Image | SAM Masking | GUI Flow | Motion Edited |

Figure A1: **Overview of GUI.** Our GUI allows a user to start with an image, mask out the objects they want to motion-edit, and construct an optical flow to be used with our method.

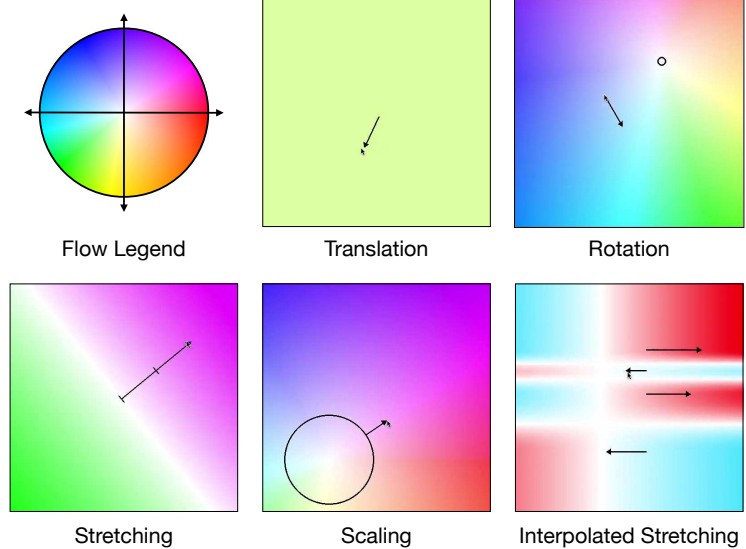

Figure A2: **Implemented GUIs.** Our GUI supports the synthesis of optical flows by clicking and dragging control arrows. We describe the interface for each of the above examples. **Translation:** Global translations are parameterized by the direction and magnitude of the arrow dragged out by the user. **Rotation:** The initial click defines the center of rotation. The angle of rotation is parameterized by the distance the user drags the cursor away from the center. **Stretching:** The initial click determines the center of stretching. The dragged arrow determines the angle at which to stretch, and the factor by which to stretch. The notch on the arrow indicates unity, where the stretching factor is equal to 1. Dragging an arrow between the center and the notch indicates squeezing, and dragging an arrow away from the notch and center indicates squeezing. **Scaling:** The center of scaling is determined by the initial click of the user. The factor by which to scale is determined by the arrow's distance from the center of scaling. The circle indicates unity, where the scaling factor is equal to 1. Dragging an arrow inside the circle equates to shrinking, and dragging an arrow outside of the circle equates to expanding. **Interpolated stretching:** We can take the "stretching" UI and allow a user to draw multiple arrows, each defining a stretch (or squeeze). Treating these arrows as "control points," we can interpolate between these stretches and squeezes to synthesize complex deformations.

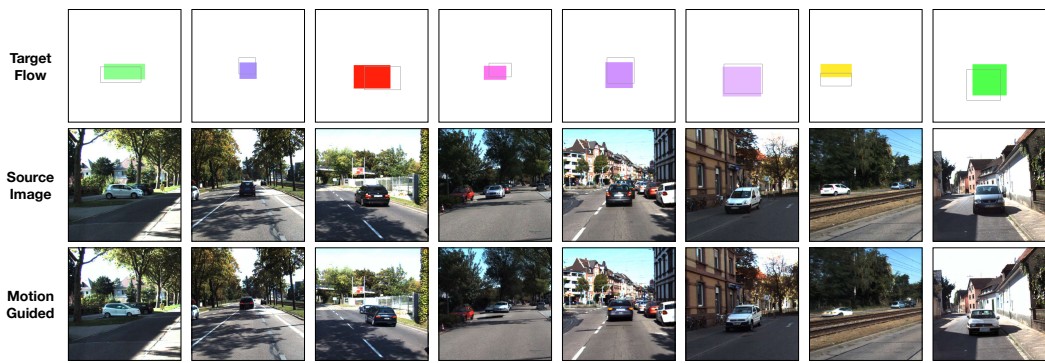

Figure A3: **KITTI Dataset Examples.** We show examples of motion guided edits on the automatically generated KITTI dataset. Two failure cases are shown on the very right. One in which the car is distorted rather than moved on to the railroad tracks, and one in which the identity of the car changes to satisfy the target flow. Because the exact motion can be difficult to interpret from the flow visualization alone, we also draw the target location of the bounding box in the top row.

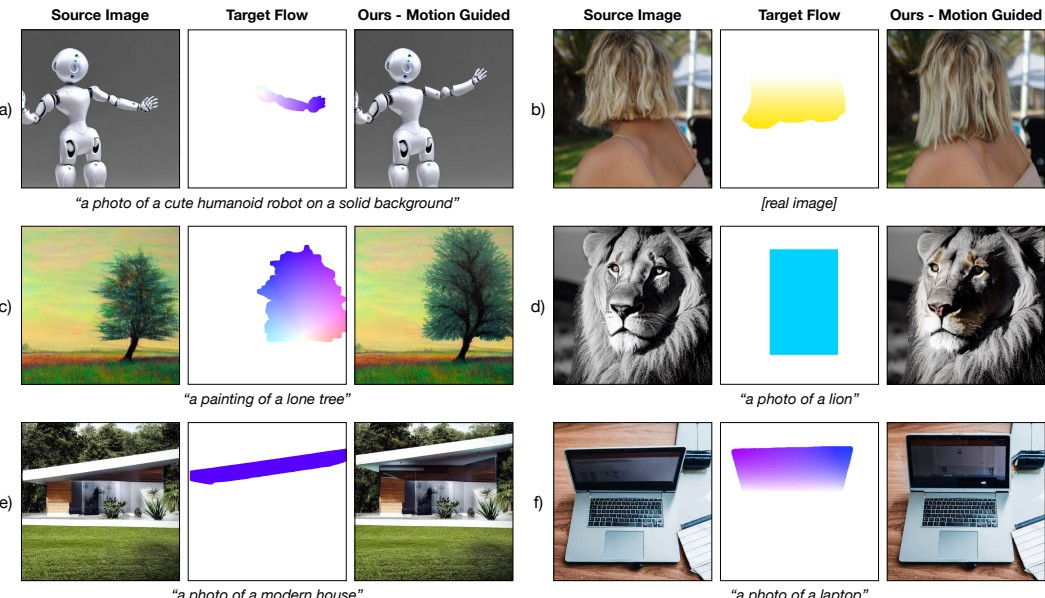

Figure A4: **Additional examples.** We present additional results using our motion guidance method. These include target flows that are (a) rotations (b) stretches (c) scaling (d, e) translations and (f) homographies.

For our curated dataset we sample images from Stable Diffusion using custom prompts and collect real images from Unsplash. We then handcraft interesting target flows for each image. For the InstructPix2Pix baselines we then manually annotate an instruction describing the motion the target flow should apply to the source image. In addition, for our evaluations we further augment the dataset by scaling each target flow by a total of 6 factors, with the aim of covering a wide range of flow magnitudes. This results in 204 examples.

## A3  ADDITIONAL RESULTS

We show more examples of motion guidance in Figure A4. In addition, we show multiple samples for source image and target flow pairs with both ranked sampling (see Appendix A1 for details) in Figure A5 and random sampling in Figure A6. We also show more qualitative examples using baseline methods and our method in Figure A7. Finally, we show samples and their flow with respect to the source image over the course of the denoising process in Figure A8.

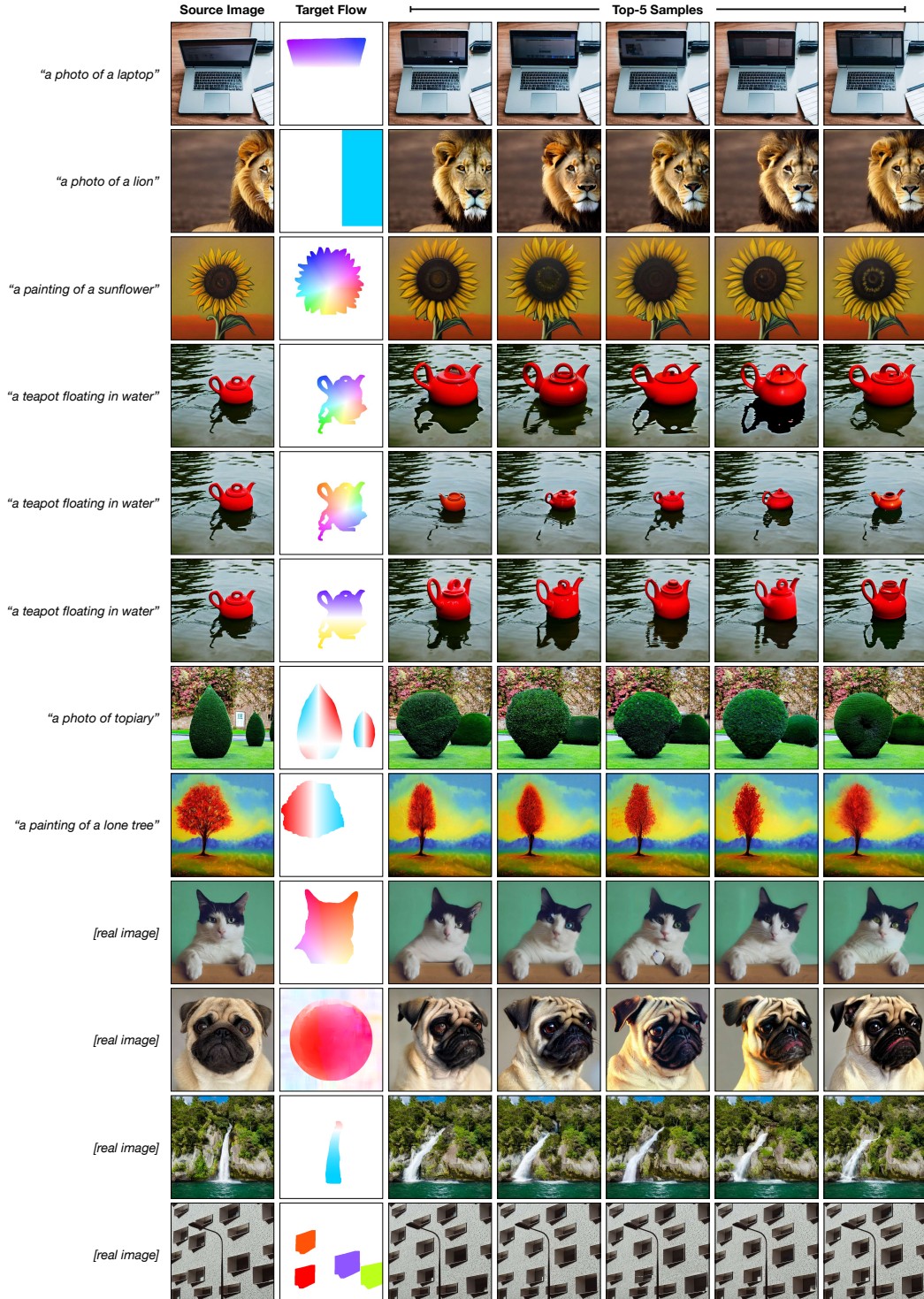

Figure A5: **Top 5 Ranked Samples.** We show a selection of the top 5 samples after reranking with our guidance loss, from a set of 32 samples. The samples are ordered left to right, better to worse ranked. For a random set of samples using the same images and target flows please see Figure A6.

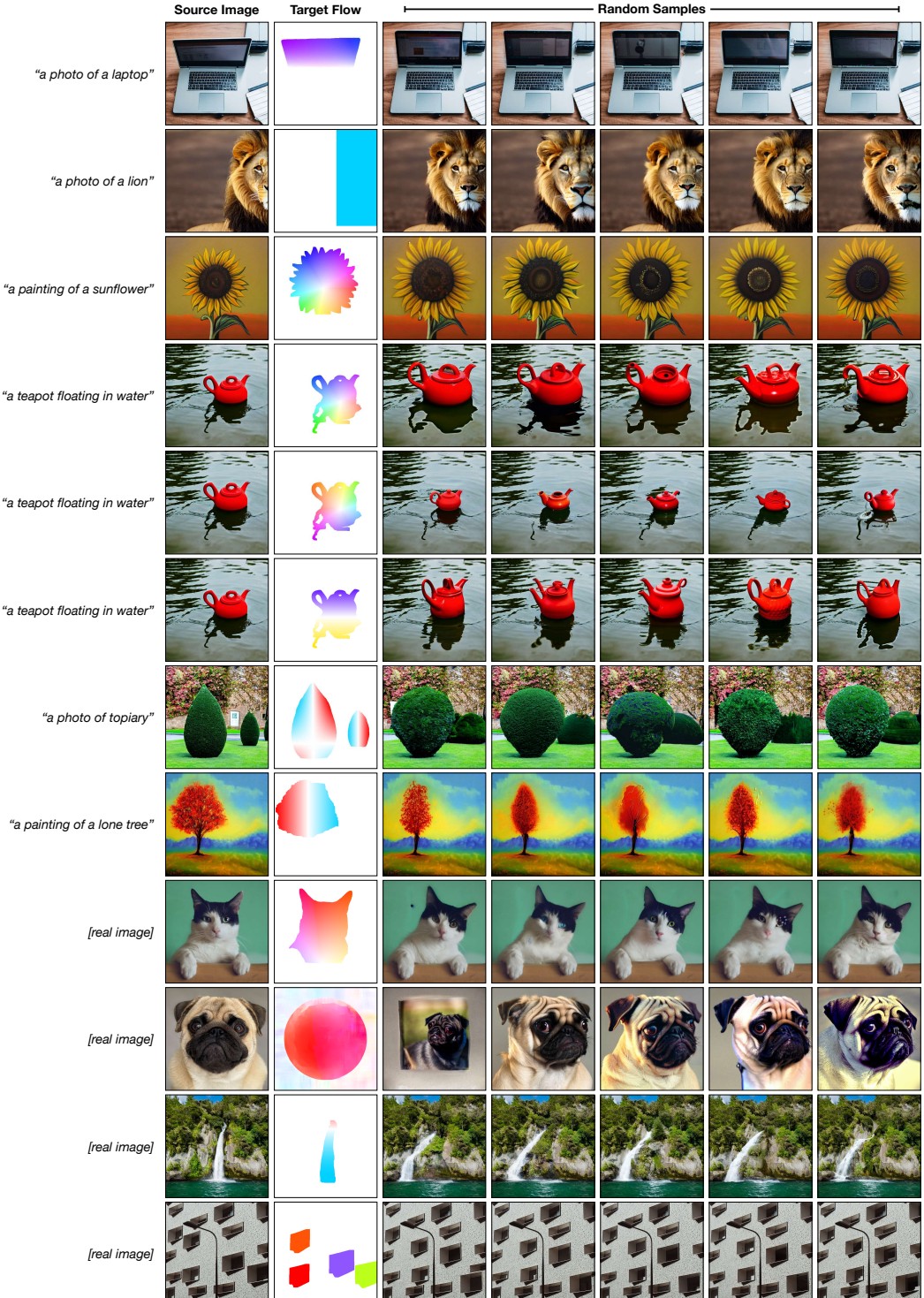

Figure A6: **Random Samples.** We show a random set of 5 samples for each image and target flow pair. These samples are of slightly lower quality than the re-ranked examples from Figure A5, as can be seen in the teapot handle, the texture of the topiary bush, the pug, or the painting of a tree.

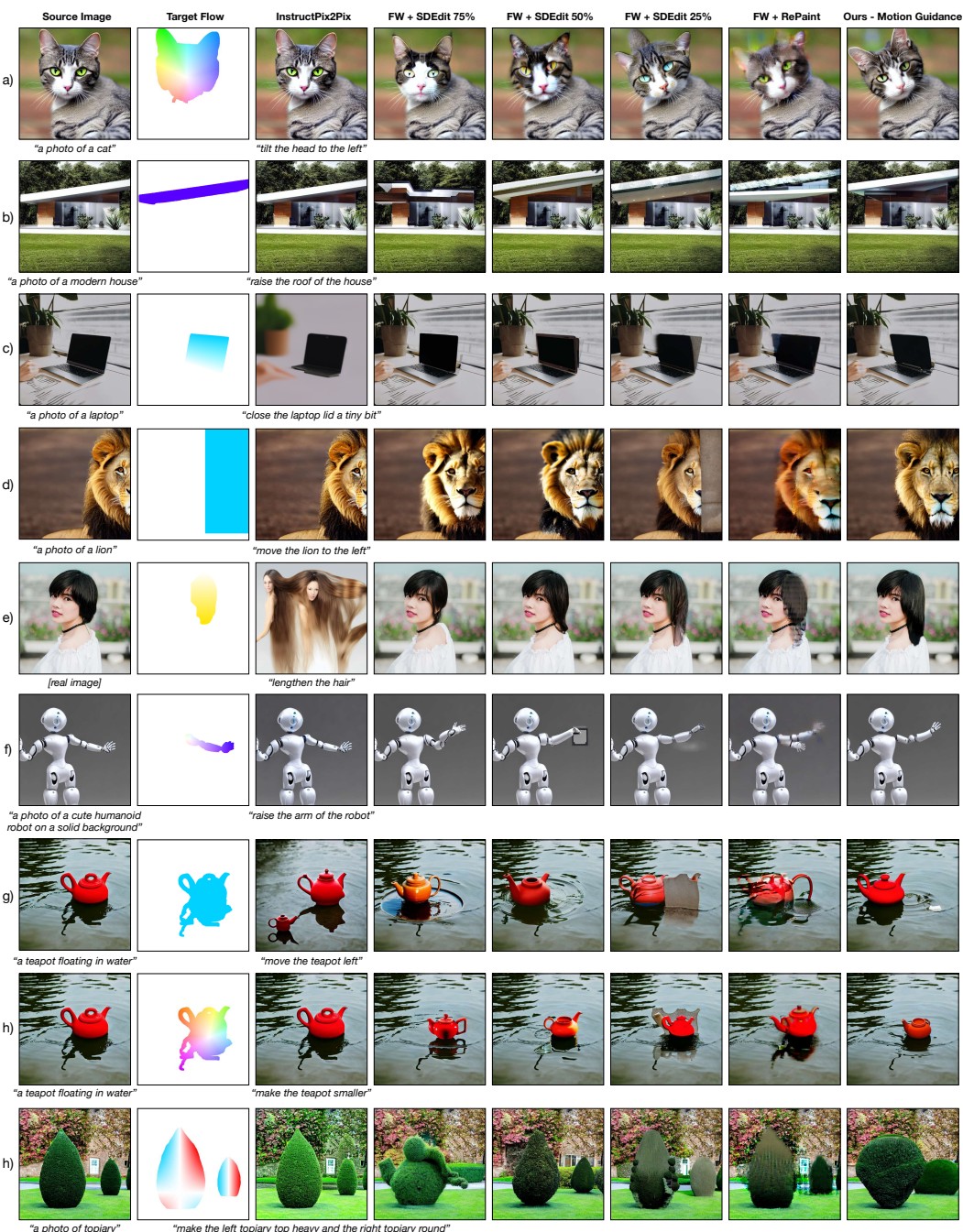

Figure A7: **Additional Baseline Images.** We qualitatively compare baselines to our method on additional examples. For further discussion please see Section 4.4.

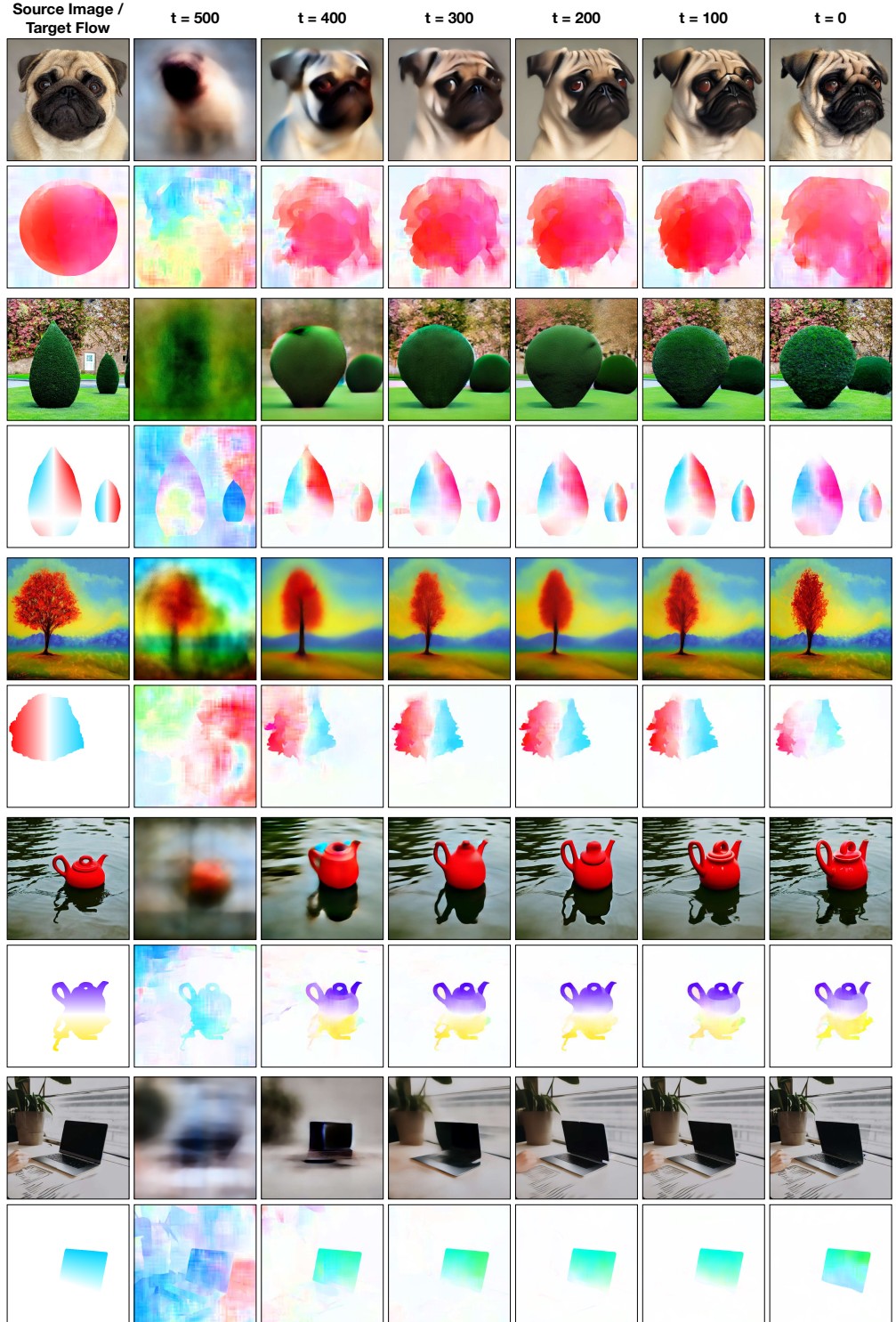

Figure A8: **Denoising Samples and Flows.** We show samples through the denoising process, along with the computed flows with respect to the source image. Specifically, we show the one-step approximation of the clean data point at timestep $t$, as well as its flow with respect to the source image as estimated by RAFT. Note that because our schedule removes guidance for the final 100 steps, which we found improved results, the final flow for $t = 0$ is not as similar to the target flow as previous timesteps.

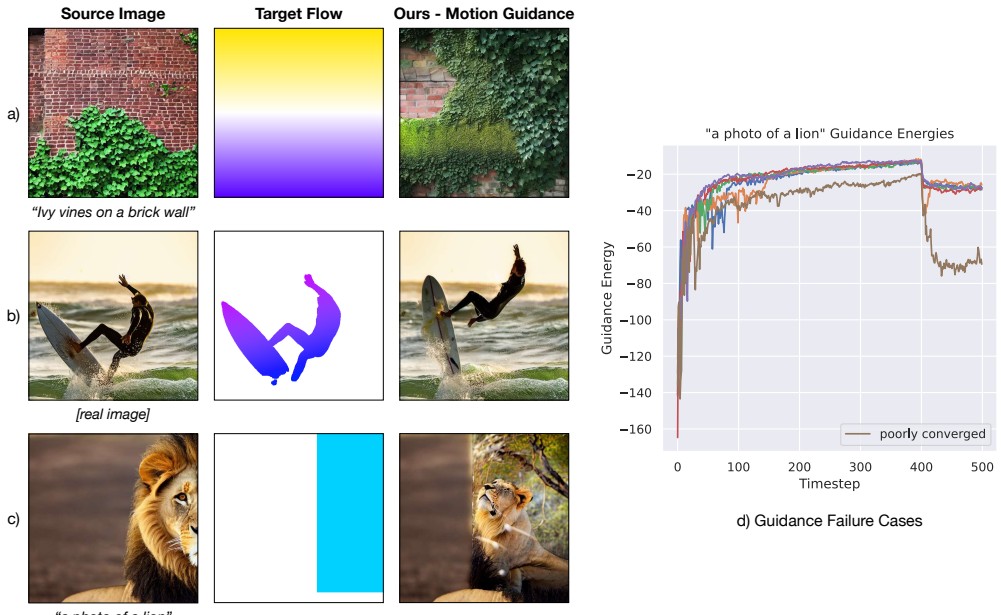

Figure A9: **Failure Cases.** We show various failure cases of our method. (a) Our method cannot handle out-of-distribution target flows, such as a vertical flip. (b) In some cases our method alters the content of the source image, such as the surfboard and the removal of the right leg of the surfer. (c) Guidance can sometimes be unstable and randomly result in poorly converged images, such as the lion example. In (d) we plot the guidance energies over the denoising process for the sample in (c) in brown, along with five other samples that successfully converged. The dip at timestep 400 is the result of our guidance schedule.

## A4 LIMITATIONS AND FAILURE CASES

Our method can effectively apply motion-based manipulations to both real and synthesized images for a wide range of target optical flows, but it also has limitations, which we discuss here.

**Target Flows** Our method works for a wide range of flows, as demonstrated in Figures 1, 2, 3, 4, 5, 7, and A4, such as compositions of translations, rotations, stretches, shears, homographies, and deformations. However, we also found that there are some target flows that do not produce good results. For example, we could not get horizontal or vertical flips to work. We hypothesize this is because these target flows are out-of-domain for off-the-shelf optical flow networks. We give an example in which we try to optimize a vertical flip target flow in Figure A9a.

**Diffusion and Guidance** Because our model is based on diffusion guidance we inherit the drawbacks of diffusion models and guidance. For one, our method is slow to sample from. Much of our overhead comes from recursive denoising, which scales linearly with sampling time. Depending on the number of recursive denoising steps, we take 15 minutes in our best case with 2 recursive steps and in the worst we take around 70 minutes with 10 recursive steps, which is on par with previous work (Bansal et al., 2023). In addition we find that guidance is sometimes unstable, which is exacerbated by the large and complex optical flow models we are backpropagating through. We show an example of guidance failure in Figure A9c and show the corresponding guidance energy over the reverse diffusion process in Figure A9d, along with the energy plots for successful samples. We hypothesize this instability comes from a trade-off on guidance strength. A large guidance strength results in a lower guidance loss, but also increases the magnitude of the perturbation, and thereby the chances that sampling may diverge.

**Forgetting and Identity** Our method is also prone to "forgetting" the content of the source image. Minor cases of this can be seen in Figure A4f, in which the content of the laptop screen changes, or in Figure 1h in which the reflection in the window slightly changes. This problem can be exacerbated by real images for which we have no caption we can condition on, as shown in Figure A9b.

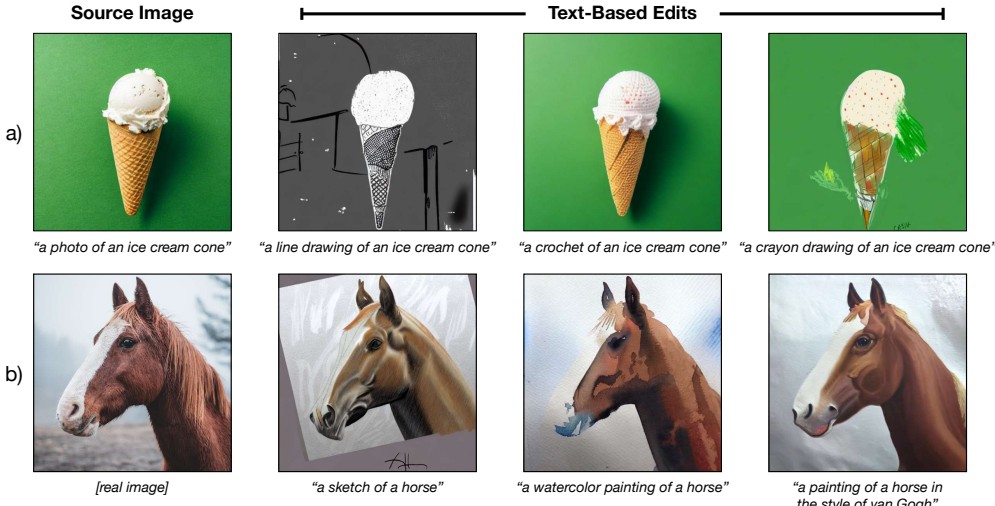

Figure A10: **Text-Conditioned Image-to-Image Translation.** Setting the target flow to zero and applying motion guidance while using a new text prompt can enable text-based edits while keeping the structure of the source image.

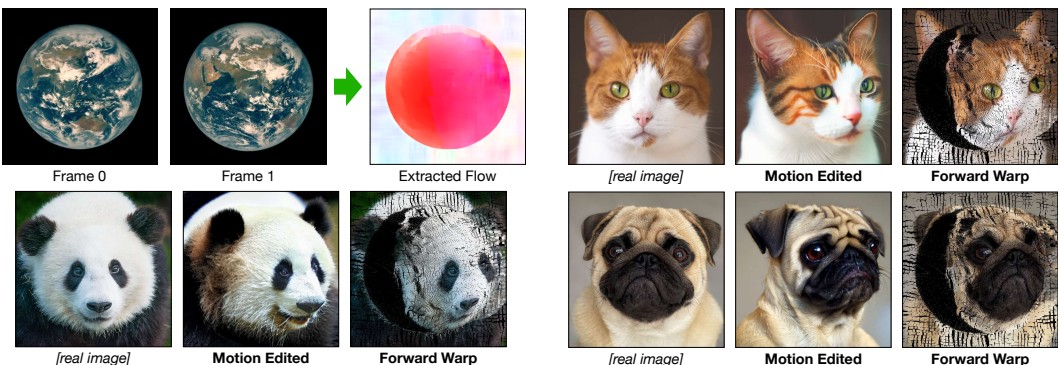

Figure A11: **Motion Transfer Compared to Forward Warp.** We show the misalignments between the target flow and the source image by forward warping.

## A5 ADDITIONAL APPLICATIONS

**Text Conditioned Image to Image Translation.** Setting the target flow to be zero everywhere allows us to modify an image while keeping its high level structure by providing an appropriate prompt, as shown in Figure A10. We note that this is not simply setting the weight on the flow loss to be zero. Rather, we are constraining the sample to have zero flow with respect to a source image. This allows us to produce quite different images that share the same structure as the source image, as in the "line drawing" in Figure A10a. While we can produce high quality results, we find that our technique is not as robust as other existing methods for text conditioned image-to-image translation such as Prompt-to-Prompt (Hertz et al., 2022). We believe this is because the flow network must compute the optical flow between two images of different styles, which is typically quite out-of-domain for the model. We hope that future progress in motion estimators may improve the robustness of this technique.

## A6 ADDITIONAL MOTION TRANSFER ANALYSIS

Motion transfer is particularly difficult because the flow does not align with the source image very well. We demonstrate this by forward warping the source images by the target flow from Figure A11. Our results do not look like the forward warped results because we optimize a soft objective: produce a likely image from the data distribution learned by the diffusion model, that simultaneously has low guidance energy.

| f = 0.1 | f = 0.2 | f = 0.3 | f = 0.4 | f = 0.5 |
|---|---|---|---|---|

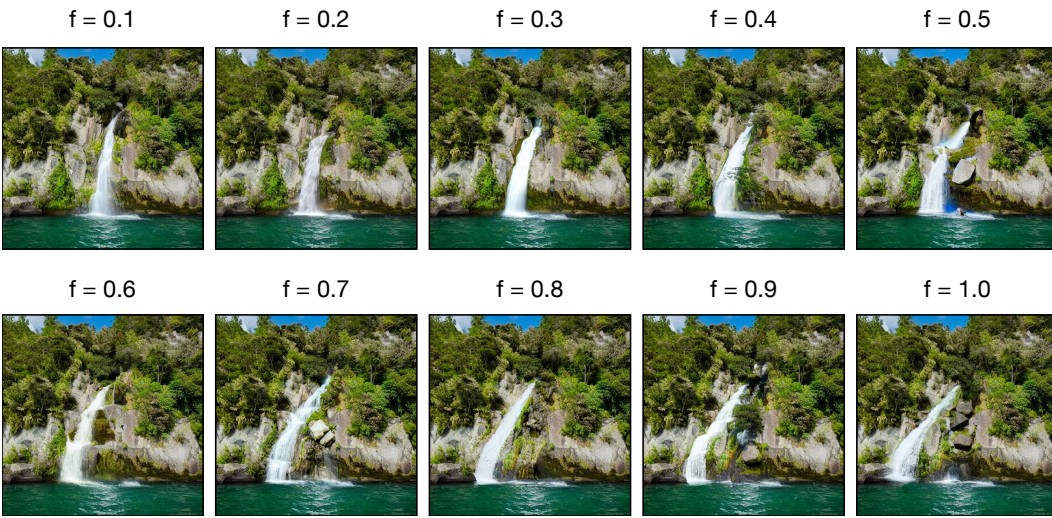

| f = 0.6 | f = 0.7 | f = 0.8 | f = 0.9 | f = 1.0 |
|---|---|---|---|---|

Figure A12: **Motion Guidance on Scaled Flows** We show motion guidance on the waterfall example from Figure 3, but we scale the flow by a factor of $f$ for each image.

## A7 VIDEOS FROM MOTION GUIDANCE

We show a sequence of motion edits on the waterfall example from Figure 3. Each sample uses the same target flow, but scaled by some factor $f$. These demonstrate our model working for more fine-grained flow than the large motions presented in the main body of the paper. In addition, this indicates the possibility of using our method to construct videos. However, we note that because the images are sampled independently, there is no coherence to how our method fills in disocclusions, resulting in a slightly choppy video. We leave this problem for future work.

