# OpenReview forum: "Motion Guidance: Diffusion-Based Image Editing with Differentiable Motion Estimators"
_ICLR.cc/2024/Conference — ICLR 2024 poster_

### Official Review · Reviewer_6Wgo · 2023-10-31

**Soundness:** 3 good
**Presentation:** 3 good
**Contribution:** 3 good
**Rating:** 6
**Confidence:** 3

**Summary:**

The authors propose a run-time image editing method based on diffusion models using an optical flow as input. The method can synthesize a realistic looking image given the reference image and optical flow fields. The synthesis is done via a new motion guidance, which is composed of a flow reconstruction loss between a provided flow map to 2D optical flow estimate from reference image to the generated image, and a color loss to ensures the generated image is consistent to input image when warping back. The authors provide extensive visual examples and ablations to show the efficacy of this approach

**Strengths:**

* The paper is technically sound and simple. The approach should be easy to implement at runtime, and can be pretty general for most image synthesis backbones.
* The authors provided very clear ablations study for many implementation details in the paper, including how to do denoising steps, editing mask, handling occlusions, which are crucial for the success. I like the provided examples which very clearly highlight the issues of each item if not being used.
* The method performs favorable compared to alternative baselines. The method provides great visual consistency with respect to the input image and flow map.

**Weaknesses:**

* As the author indicated in Sec 4.6, it is hard to create pixel-wise flow map by hand, which I personally will be the biggest limitation for this method to be applied in real-world use-case. I am not exactly how the authors demonstrate all the examples in results section, but from the results, they definitely look a real pixel aligned motion field regarding the input image, which is almost impossible to get in applications. From the motion transfer results, the authors discuss using "fewer recursive denoising steps" to get slightly better results. I wonder that's related to the gap in pixel aligned optical flow. I think it worth providing a careful analysis here to help readers understand the potential gap and insights to address these issues which is how this method used in real-world use-cases. I wonder whether there are also tasks (e.g. video stylization) that can better map the input of this method.
* A suggestion, not a weakness. The demonstration of flow map can be more clear if provided with the color mapping chat with the flow (motion to color).

**Questions:**

* For the recusive denoising, is the K constant number for all the experiments? I did not find what the parameter is used in the implementation details section.
* In flow loss discussion on page 6, the authors mentions "Without the flow loss our method is able to move objects to the correct location but often hallucinates things in disoccluded areas". Though it is true from the fig.3, the disocclusion regions looks quite wrong with flow loss, I did not quite understand why it is the case. The disocclusion region should not really impact the optical flow since they don't really have correspondence. One possible explanation is that they could potentially create ambiguity between the input image and synthesize image for the estimated flow, and adding flow floss could drive it to remove that potential ambiguity. I wonder whether that's the potential case?
* Both Eq. (2) & (3) seems require the gradient w.r.t. the whole flow estimator if I understand correctly. That could be really slow? I wonder the compute time for each. For eq (3), will it make any difference if using f instead of F(x^{\star}, x) ?

---

> ### Author Response · Authors · 2023-11-21
> **Response to Reviewer 6Wgo**
>
> We thank the reviewer for their positive and comprehensive feedback. Below are our responses.
>
> &nbsp;
>
> **Target Flow Construction**
>
> Generating a target flow field may seem complicated but is actually rather straightforward, as outlined in Appendix A1. All flows in our paper (except for the motion transfer results, which are extracted from real videos), are compositions of simple, elementary flows.  As such, it is relatively straightforward to imagine a UI, similar to existing software such as MS Paint, Keynote, or Illustrator, which allows a user to “paint” a target flow.
>
> Additionally, in Appendix A1 we describe how we create target flows that are pixel-aligned with the input image by using SAM [Kirillov et al. 2023], a simple, fast, and highly user-friendly, off-the-shelf segmentation network.
>
> We also agree with the reviewer that some flows are hard to construct, which is why we include the motion transfer results. We show that these flows can be obtained by extracting them from existing video.
>
> &nbsp;
>
> **Recursive Steps in Motion Transfer Results**
>
> We agree that fewer recursive steps being needed for the motion transfer experiments may have something to do with the pixel misalignment. Our hypothesis is that fewer recursive steps helps in this case because we do not want to optimize our guidance loss function too aggressively, given that our flow is somewhat misaligned. This is because we are effectively taking fewer “gradient steps” on our guidance function. We provide a visualization of the extent of misalignment between the extracted motion and the source images in Appendix Section A6 and Figure A9. Overall, motion transfer works because our method optimizes a soft objective, which is controlled partly through the number of recursive steps.
>
> &nbsp;
>
> **Flow Legend**
>
> We have already included a flow legend in the form of a colorwheel in Figure 1, where we show how the hue and value of a color corresponds to the motion we are trying to depict. We opted not to include this legend in each figure due to space considerations, but we have updated the caption of each figure to refer back to this legend to clear up any confusion. We thank the reviewer for pointing this out.
>
> &nbsp;
>
> **Recursive Denoising Value**
>
> Information on hyperparameters can be found in section A1, “Implementation Details,” in the appendix. We use a value of K=10 for all experiments except for the motion transfer experiments, in which we use a value of 10, 5, and 2 for the cat, pug, and panda respectively. We realize we did not specify the exact parameters for the motion transfer experiments, and we have updated Appendix Section A1 accordingly. We thank the reviewer for spotting this.
>
> &nbsp;
>
> **Role of the Flow Loss**
>
> The reason why removing the flow loss results in hallucinations is because the color loss provides no signal to disoccluded regions, but the diffusion network still denoises in the disoccluded areas. There is nothing preventing the diffusion model from hallucinating what it wants in those areas, in particular, from hallucinating an extra apple or an additional waterfall. We also believe the reviewer’s hypothesis for why the flow loss helps. By adding the additional constraint we encourage the network to remove any potential ambiguity in the flow.
>
> &nbsp;
>
> **Efficiency of Flow Backpropagation**
>
> Backpropagating through the entire loss function takes up roughly 50% of the total sampling time. Using the target flow in the color loss to circumvent the backpropagation overhead is an interesting idea, but does not work because we have to backpropagate through the flow loss anyways. We do note this trick is still useful, and we use it in our ablations when we try to completely remove the flow loss from our guidance loss.
>
> &nbsp;
>
> We hope these comments have addressed the reviewer’s questions. We would like to thank them for their time, and ask that they consider raising their score for our paper.

---

> > ### Comment · Reviewer_6Wgo · 2023-11-23
> >
> > Thanks for the comments addressing my questions. I did not notice the appendix 1 which describes the motion generation. I think that's clear to me.
> >
> > > As such, it is relatively straightforward to imagine a UI, similar to existing software such as MS Paint, Keynote, or Illustrator, which allows a user to “paint” a target flow.
> >
> > I will suggest the authors to create qualitative examples to illustrate this clearly to readers. Apparently the usability of such method in downstream application is a concern highlighted by all reviewers.
> >
> > I will tend to remain my initial ratings above the line for acceptance.

---

### Official Review · Reviewer_wHkp · 2023-10-31

**Soundness:** 3 good
**Presentation:** 3 good
**Contribution:** 3 good
**Rating:** 6
**Confidence:** 4

**Summary:**

This paper presents a new method for editing images with diffusion models and optical flow guidance. The proposed method does not require any network training or finetuning, and does not modified the diffusion network architecture. Instead, it applies an additional loss to guide the diffusion sampling process, making the edited image follow motion guidance without changing image contents. Experiments show that the proposed method can handle various type of motion guidance and generate high-quality results.

**Strengths:**

* The core of the proposed image editing method is a loss function that incorporates a pretrained optical flow network for motion supervision, and does not need any network training or finetuning. Therefore, it requires no data collection or network modification.

* The proposed method is simple yet flexible and supports a wide range of motion guidance. When manipulating the image contents, the proposed method is able to hallucinate missing pixels like disoccluded areas or backgrounds.  The qualitative results are impressive.

* The authors conduct extensive experiments to demonstrate the effectiveness and flexibility of the proposed method. The experiments are solid and convincing.

* The paper is overall well-written and easy to follow.

**Weaknesses:**

* The proposed method requires dense optical flows as the supervision signal. This is not a user-friendly input, as it is not clear how to obtain such flows without programming. DragGAN [Pan et al. 2023] is more user-friendly; users just need to simply drag a small number of pixels to manipulate images. In addition,  DragGAN provide feedback to users almost immediately, while the proposed method is slow to execute.

* In Figure 5, the authors conduct a comparison experiment with DragGAN on out-of-domain images. However, StyleGANs are trained on narrow domains and DragGAN works better if the input image falls in the training image domain.  Although GAN inversion still works for out-of-domain images, I think it would be better if the authors can compare with DragGAN using in-domain images (e.g., human facial images) to make the experiment stronger and more convincing.

**Questions:**

* I think the proposed method can be easily extended for video generation by providing a sequence of flow guidance. It would make the paper stronger if the authors could provide such an example.

---

> ### Author Response · Authors · 2023-11-21
> **Response to Reviewer wHkp**
>
> We thank the reviewer for their positive and comprehensive feedback. Below are our responses.
>
> &nbsp;
>
> **User Interface**
>
> Generating a target flow field may seem complicated but is actually rather straightforward, as outlined in Appendix A1. All flows in our paper (except for the motion transfer results, which are extracted from real videos), are compositions of simple, elementary flows. As such, it is relatively straightforward to imagine a UI, similar to existing software such as MS Paint, Keynote, or Illustrator, which allows a user to “paint” a target flow.
>
> &nbsp;
>
> **DragGAN**
>
> We see the relationship between our method and DragGAN as a series of tradeoffs. While DragGAN can work for sparse point displacements, they sacrifice precision and control. Our method on the other hand works with dense flows, which can be much more accurate. DragGAN is quite fast, but the tradeoff is that GANs are trained only on narrow datasets, while our method can handle a far wider breadth of images _and_ can be controlled by text prompting.
>
> &nbsp;
>
> **Out of Domain DragGAN Results**
>
> The goal of Figure 5 was to illustrate the limitations of DragGAN and how our method can overcome them. We wanted to point out two things. Firstly, that DragGAN only works on narrow domains. And secondly, that for DragGAN to work on real images, it also requires a very good GAN inversion method. This is discussed in section 4.4.
>
> &nbsp;
>
> **Video Generation**
>
> We thank the reviewer for this idea. We have updated the appendix in the supplemental material to add a sequence of deformations by increasingly larger target flows, which can be found in Figure A10 and Section A7. While the motion edits look good, the images are sampled independently of each other. Because of this the model fills in disocclusions inconsistently, and the frames form a slightly choppy video. We believe more work must be done to address this issue, and leave this for future work.
>
> &nbsp;
>
> We hope these comments have addressed the reviewer’s questions. We would like to thank them for their time, and ask that they consider raising their score for our paper.

---

### Official Review · Reviewer_iTDk · 2023-10-31

**Soundness:** 4 excellent
**Presentation:** 4 excellent
**Contribution:** 4 excellent
**Rating:** 8
**Confidence:** 4

**Summary:**

This paper proposes a new motion guidance technique for diffusion models. The specified motion field can be arbitrary, but the reconstructed images are always plausible thanks to the diffusion model and the color loss. The range of supported motions is diverse, starting from simple translation and up to complex non-rigid motions. The method is extensively evaluated on several datasets, both qualitatively and quantitatively.

**Strengths:**

- An extensive related work section with good literature overview.
- The method supports many different types of transformations: translation, rotation, scaling, stretching, and other complex deformations.
- The method does not require any additional training and runs directly on the required inputs. It is also designed in such a way that it is independent on the used architecture.
- The results look visually pleasing.
- Many qualitative ablation studies support main contributions.
- The paper is well-written and easy to understand.

**Weaknesses:**

- Unfortunately, all experiments and ablations are done mostly qualitatively. This brings a question whether the results are cherry-picked. I'd like to see more quantitative evaluations. Maybe the authors could evaluate the consistency of predictions over longer videos.
- In all experiments, the authors used optical flow (probably from a video). However, how can this method be used if the end user has only one image and wants to edit it? There is one experiment (Fig. 7), which shows that it can be estimated from a different video (and I appreciate it). However, I'd like to see if a hand-drawn optical flow could be used.
- eq. (3): later in Sec. 3.3 it is written that the color loss is masked out in "the occluded regions". However, this is not represented in eq. (3). Please update the equation to be precise.

**Questions:**

Please see the Weaknesses section.

---

> ### Author Response · Authors · 2023-11-21
> **Response to Reviewer iTDk**
>
> We thank the reviewer for their positive and comprehensive feedback. Below are our responses.
>
> &nbsp;
>
> **Quantitative Results**
>
> We first point out that we quantify the trade-off between CLIP alignment scores and Flow Loss for two different datasets, and two different flow methods in Figure 6. The majority of our results are qualitative because there is no standard benchmark for the task that we consider, and we believe that qualitative results effectively convey both the use cases of our method and the high quality of the samples from our method.
>
> &nbsp;
>
> **Random Samples**
>
> We have already provided completely random samples using our method in the appendix, in Section A3 and Figure A4. In total, there are 60 images sampled with a set random seed, all of which are of high quality in terms of image content and adherence to the target flow.
>
> &nbsp;
>
> **Hand-Drawn Optical Flow**
>
> All experiments in this paper use “hand-drawn” optical flow, which is described in Appendix section A1. These flows are compositions of simple, elementary flows, making it relatively easy to generate by hand. The only experiments for which we use optical flow from a video is the “motion transfer” results in Figure 7.
>
> &nbsp;
>
> **Guidance Function Equation**
>
> We thank the reviewer for catching this. We have updated equation 4 to be more precise and have included a short explanation.
>
> &nbsp;
>
> We hope these comments have addressed the reviewer’s questions, and we would like to thank them for their time.

---

### Official Review · Reviewer_wQXZ · 2023-11-04

**Soundness:** 3 good
**Presentation:** 3 good
**Contribution:** 2 fair
**Rating:** 8
**Confidence:** 4

**Summary:**

This paper proposes a way to edit images using by applying a desired motion field. This is done by guiding a pretrained diffusion model at inference time according to a user-specified advection field. The approach is similar to classifier guidance---a differentiable guidance loss utilizes an off-the-shelf flow prediction deep network, encouraging the flow from the source image to the generated image to resemble the target flow at each diffusion step. The authors compare to several GAN-- and diffusion--based approaches.

**Strengths:**

Being able to inject user control into the output of a diffusion model is an important goal. Using a dense advection field as guidance is an intuitive way to manipulate images and could be potentially be useful for enforcing other spatial constraints on diffusion model outputs like temporal consistency and so on. The paper is clearly written, and a comprehensive ablation study validates many of the design choices.

**Weaknesses:**

My main concern has to do with lack of comparisons with recent works. While the authors discuss some of these in related works, they do not share any qualitative or quantitative comparisons. In particular, [Shi et al. 2023] and [Mou et al. 2023] seem to enable very similar functionality. While I recognize the authors' explanation that their approach is more efficient in that it requires no additional training or fine-tuning and is only done at inference time, I think it would be necessary see how these other methods perform on the same examples.

I would also be interested in seeing how this approach handles more complicated, higher-frequency flow fields. It seems like one big advantage of being able to control a diffusion model via a dense flow field is to enforce fine-grained motion. However, all the examples feature quite coarse and global transformations.

Finally, I am a little concerned about possible hallucination or identity loss resulting in these transformations. This is particularly evident in the motion transfer examples of Figure 7---as a result of the of the motion edit, the subject in the image also changes considerably. Is there a way to mitigate this effect?

**Questions:**

What is the intuition for why the edit mask is necessary to make this approach work? Shouldn't the fact that the flow field is identity on the unchanged regions be sufficient?

I would be curious to see what the one step approximation derived by equation (5) looks like as well as the results of passing it through the flow estimation model rather than the final denoised image. Is it a sufficiently good approximation and not out of distribution?

Does using a pretrained neural flow prediction model offer a significant  advantage over differentiating through an actual optical flow algorithm? For instance, works like "Supervision-by-Registration" [Dong et al. 2018] have done this for other tasks---I'm wondering how well a similar approach would work here.

---------

Post-rebuttal:

Thank you to the authors for their clarifications. I did not realize that [Shi et al. 2023] and [Mou et al. 2023] are concurrent work. Given this fact and the other addressed points, I have updated my score and recommend acceptance.

---

> ### Author Response · Authors · 2023-11-21
> **Response to Reviewer wQXZ**
>
> We thank the reviewer for their constructive and comprehensive feedback. Below are our responses.
>
> &nbsp;
>
> **Comparison to Recent Works**
>
> [Per ICLR reviewer guidelines](https://iclr.cc/Conferences/2024/ReviewerGuide), authors are not expected to compare with work **published within four months of the main paper deadline (Sept. 28th, 2023).** Not only are [Shi et al. 2023] and [Mou et al. 2023] **not even published**, their preprints were only just uploaded to arXiv within this four month period. In fact, it appears that one of these works [may have been submitted to this conference as well](https://openreview.net/forum?id=wj7nvRqdp8), clearly making it concurrent work.
>
> Our paper does, however, contain a comparison to recent related work, DragGAN, even though we were not required to since it [published on July 2023](https://dl.acm.org/doi/abs/10.1145/3588432.3591500). This is because we thought the reader would benefit from such a comparison, and because we thought we could make the comparison fairly since DragGAN had released a finalized version of their code. Overall, we hope this conveys to the reviewer that we have made a good faith effort to compare our method to relevant baselines.
>
> If the reviewer has any additional questions or comments, we would love to discuss further. **However, we believe we have addressed the reviewer’s main concern, and we would like to ask the reviewer to consider improving their rating.**
>
> &nbsp;
>
> **Complex Flow Fields**
>
> We disagree with the assessment that our flow fields are “coarse and global,” especially when compared to prior work in this area, which handle only sparse point-wise motion edits. For example, the deformation of the topiary in Figure 1, the waterfall in Figure 3, the homography on the laptop lid in Figure 3, and the river in Figure 4 are all quite complex. Each of these flow fields consists of continuously varying, highly non-global, fine-grained displacements. This is not to mention the rotations and scaling flow fields, which while conceptually very simple, are quite mechanically complex and require fine-grain control of the content to achieve.
>
> &nbsp;
>
> **Hallucinations in Motion Transfer**
>
> First, we would like to note that our method does not hallucinate for the vast majority of our examples, as seen in Figures 1, 2, 3, 4, and 5. We believe that “motion transfer” is an extreme case, and results in small changes in lighting or subject identity because the extracted optical flow does not perfectly match the source image. We add a demonstration of this in Figure A9 of the appendix, with discussion in Section A6. Because of this mismatch, the diffusion model must do significant work in hallucinating some of the missing or unrealistic detail, and “correct for” the target flow. This is one of the benefits of our soft optimization method. In fact, to some extent objects *must* be hallucinated. For example, naively stretching the topiary in Figure 1 would result in artifacts in the leaves. That the diffusion model “hallucinates” details can make the edit much more realistic.
>
> &nbsp;
>
> **Edit Mask**
>
> Our method does not necessarily need an edit mask to work. For example see the motion transfer results in Figure 7. We add it for two reasons. One is because we wanted to mimic the edit masks of other methods such as [Pan et al. 2023], which seems useful. And the other is that guidance is actually a soft constraint. It is akin to doing gradient descent on two different objective functions. Therefore, even by specifying zero flow in an area, we are not guaranteed zero flow in that area. This is quite useful sometimes, as we can follow the “intention” of the target flow while not being bound to its exact form (as discussed above with respect to motion transfer). But it also means that an edit mask can help to make this constraint hard.
>
> &nbsp;
>
> **One Step Approximation and Flow Estimates**
>
> We have already provided examples of one step approximations and corresponding flow estimates in Figure A6 in the appendix. As can be seen, the one step approximations are quite good, even in earlier timesteps. This allows our guidance function to control the optical flow throughout the entire denoising trajectory.
>
> &nbsp;
>
> **Differentiating Through Classical Flow Methods**
>
> We choose to use neural network based flow methods because they are state of the art. We choose RAFT specifically because it is a very high quality, fully-differentiable optical flow method, and has been shown to work in diverse scenarios. This allows us to make motion-based edits on a diverse set of source images with a broad range of target flows.
>
> &nbsp;
>
> We hope these comments have addressed the reviewer’s questions. We would like to thank them for their time, and again ask that they consider raising their score for our paper.

---

### Meta-Review · Area_Chair_tk9C · 2023-12-09

**Metareview:**

All the reviewers are positive about the paper: the use of a dense motion field for image manipulation is intuitive and potentially useful for spatial constraints in diffusion models, the paper is well written and clear, the method supports complex and versatile transformations without extra training, and the qualitative results are visually pleasing supporting the main contribution. On the other hand, they also shared the weaknesses of the paper: it lacks qualitative and quantitative comparisons with recent work, whether it can handle more complex, higher-frequency flow fields and fine-grained motion, there are concerns about hallucination or identity loss in transformation (as shown in the motion transfer examples), and the requirement for dense flow field as supervision is not user-friendly.

The authors addressed these concerns carefully in the rebuttal and the paper received two 6 and two 8 ratings. The AC recommends accept. Please incorporate many important rebuttal points into the final version.

The AC also sees the limiting factor of using flow field for image manipulation in the paper, so the acquisition of the flow field should be fully discussed in the beginning of the paper. Some of the flow fields are derived from simply transformation of the foreground objects, while others are inferred from a reference sequence. It’s better to make it clear up in the front.

**Justification For Why Not Higher Score:**

Flow field is a great intermediate representation for image editing but not so great as the interface for users. User-friendliness is a very important factor for the impact of this paper. Besides, fidelity of the transfer is of concerns especially when the paper is framed under "flow field", so the expectation is set up to be exact match.

**Justification For Why Not Lower Score:**

All four reviewers are positive about the paper.

---

### Decision · Program_Chairs · 2024-01-16

Accept (poster)